# Geometric Self-Supervised Pretraining on 3D Protein Structures using Subgraphs

## Abstract

Protein representation learning aims to learn informative protein embeddings capable of addressing crucial biological questions, such as protein function prediction. Although sequence-based transformer models have shown promising results by leveraging the vast amount of protein sequence data in a self-supervised way, there is still a gap in exploiting the available 3D protein structures. In this work, we propose a pre-training scheme going beyond trivial masking methods leveraging 3D and hierarchical structures of proteins. We propose a novel self-supervised method to pretrain 3D graph neural networks on 3D protein structures, by predicting the distances between local geometric centroids of protein subgraphs and the global geometric centroid of the protein. By considering subgraphs and their relationships to the global protein structure, our model can better learn the geometric properties of the protein structure. We experimentally show that our proposed pertaining strategy leads to significant improvements up to 6%, in the performance of 3D GNNs in various protein classification tasks. Our work opens new possibilities in unsupervised learning for protein graph models while eliminating the need for multiple views, augmentations, or masking strategies which are currently used so far.

## 1 Introduction

Proteins are fundamental biological macromolecules, responsible for a variety of functions within living organisms, ranging from catalyzing metabolic reactions, DNA replication, and signal transduction, to providing structural support in cells and tissues (Conrado et al., 2008; Whitford, 2013; Tye, 1999). Predicting protein function is one of the most important problems in bioinformatics, with extensive applications in drug design, drug discovery and disease modeling (Skolnick & Brylinski, 2009; Luo et al., 2021; Rezaei et al., 2020). However, the complexity and variability of proteins pose significant challenges for computational prediction models (Radivojac et al., 2013; Schauperl & Denny, 2022). The function of a protein is affected by its three-dimensional structure, often dictating its interactions with other molecules (Ivanisenko et al., 2005). The 3D structure of proteins provides critical knowledge that is often much harder to derive from their 1D amino acid sequences alone. Therefore, understanding and predicting protein function based purely on sequence data can be challenging without considering the 3D structural modality (Gligorijević et al., 2021; Ingraham et al., 2019).

In recent years, the advent of 3D graph neural networks (GNNs) has introduced a big potential for protein representation learning. These models utilize the graph structure of proteins, where nodes represent atoms or residues, and edges represent the bonds or spatial relationships between them (Wang et al., 2023; Zhang et al., 2022). GNNs are particularly good at processing the non-Euclidean data represented by 3D protein structures, enabling them to learn complex patterns that affect protein functionality (Swenson et al., 2020; Abdine et al., 2024).

Despite these advancements, a significant limitation remains in the field: the *absence of a unified approach to effectively leverage unlabeled 3D structures for pretraining deep learning models*. Most current methods depend heavily on labeled data, which is scarce and expensive to produce. In contrast with transformer models, which have effectively used token masking as a pretraining strategy and achieved significant success

in various fields (Vaswani et al., 2017), graph models still lack a definitive, universally accepted pretraining approach (Sun et al., 2022). Particularly for 3D structures, graph-based models face challenges in leveraging the extensive, unlabeled data available, while also struggling to manage computational demands efficiently. Most prominent approaches mask node attributes or edges and then try to predict them (Hu et al., 2020). However, they do not take into account the hierarchical structure of proteins and the important substructures or motifs that affect their function.

Our approach tackles these challenges by introducing a novel pretraining strategy for 3D GNNs, capitalizing on the geometric properties of protein structures. Specifically, we predict the Euclidean distances between the geometric centers of various protein subgraphs and the protein's overall geometric center. This method offers several advantages. First, by utilizing subgraph representations, the model can accurately learn and capture hierarchical patterns within the 3D structure. Second, it captures the relative distances between subgraphs, a valuable feature as some tasks require focusing on surface nodes, while others may need attention on more central nodes. This flexibility increases the model's ability to handle different types of protein-related tasks effectively.

The goal of our pretraining is to capture meaningful structural information about proteins that can later be fine-tuned for specific downstream tasks. By designing a pretraining task that focuses on subgraph distances, we hypothesize that our model will develop a deeper understanding of protein geometry, especially compared to simpler tasks like edge distance prediction. The intuition is that subgraph distance prediction forces the model to learn more complex interactions within the protein structure, making it a richer and more informative pretraining task.

We evaluate our approach, by pretraining various models with different featurization schemes, for protein structures, in a large amount of 3D structures from AlphaFold database (Varadi et al., 2022). We demonstrate increased performance in multiple protein classification tasks for different base architectures. Our pretraining strategy is designed to be general and adaptable, as it can be used with any model architecture that can encode the protein 3D structure. We believe our approach will lead the way and inspire more geometric self-supervised methods on 3D protein structures.

Our contributions can be summarized as follows:

- We present a new pretraining task for protein representation learning that focuses on predicting geometric distances between subgraphs. Our work presents a significant shift from traditional pretraining masking tasks, and open a new avenue in geometric self-supervised learning.

- Our proposed pretraining strategy allows the model to capture rich geometric and structural features of proteins, while maintaining a low computational overhead.

- We conduct a thorough evaluation of the proposed pretraining task using various featurization schemes and backbone models. Our results show that the proposed pretraining task consistently improves the downstream performance.

- We analyze the performance in the pretraining task and identify correlation with downstream task performance, consistent with findings in other fields, such as language modeling.

- We release the full source code and integrate our model into the ProteinWorkshop library (Jamasb et al., 2024), providing the community with tools to easily reproduce our results and extend the work for future research in protein representation learning.

## 2 Related Work

**GNNs.** Graph Neural Networks were introduced years ago (Scarselli et al., 2008), but it wasnt until the rise of deep learning that they started gaining widespread attention (Kipf & Welling, 2016; Hamilton et al., 2017; Veličković et al., 2017). Despite their variations, these models can be unified under the framework of Message Passing Neural Networks (MPNNs) (Gilmer et al., 2017). MPNNs use an iterative message

passing mechanism, where each node updates its representation by receiving messages from its neighbors. The final graph representation is obtained using a permutation invariant pooling function over the node representations. Several models have been developed to handle various types of graph structures, including those designed for heterogeneous graphs (Yu et al., 2020; Lv et al., 2021), signed graphs (Huang et al., 2021; 2019), and 3D geometric graphs (Gasteiger et al., 2020; Schütt et al., 2018; Coors et al., 2018; Du et al., 2024).

**Protein Representation Learning.** Protein representation learning aims to learn informative embeddings that capture the biological and functional characteristics of proteins. Early methods primarily focused on sequence-based representations (Kulmanov & Hoehndorf, 2020; Liu, 2017). Recent advancements have shifted towards multimodality, by integrating the structural information of proteins. For instance, methods like HoloProt (Somnath et al., 2021) incorporate sequence, surface and structure information, DeepFRI (Gligorijević et al., 2021) propose a GCN to solve protein function prediction tasks while GAT-GO (Lai & Xu, 2022) introduces an attention-based graph model. Moreover, with the advance of language models, recent models have started to integrate and encode also text information for proteins such as Prot2Text (Abdine et al., 2024), ProtST (Xu et al., 2023) and ProteinDT (Liu et al., 2023). 3D GNNs have also emerged as a promising approach to capture the spatial relationships within protein structures. Wang et al. (2023) introduced ProNet, a 3D GNN model that integrates spatial and geometric information for protein classification tasks. Schütt et al. (2018) developed SchNet, which incorporates radial basis functions to handle pairwise distances in molecular graphs. Coors et al. (2018) proposed SphereNet, a spherical representation of molecular structures that enhances spatial encoding. Our work is orthogonal to these methods, as it can be applied to various backbone architecture, aiming to improve the learned representations by leveraging the geometric structure in 3D protein data throughr pretraining.

**Graph Pretraining.** Pretraining techniques for GNNs have focused on various strategies to utilize unlabeled data effectively. Traditional methods include node and edge masking, where attributes are hidden, and the model learns to predict them (Hu et al., 2019; Xie et al., 2022). However, these methods often fail to capture the complex hierarchical and spatial patterns present in 3D structures. In contrast, our approach aim to leverage the geometric properties of 3D protein structures using different motifs, offering a novel approach to pretraining in this domain.

Graph contrastive learning methods have also gained traction as effective approaches for pretraining graph models. These methods aim to learn meaningful embeddings by contrasting different views or augmentations of the same graph, such as through node perturbations or subgraph extractions. GraphCL (You et al., 2020), which applies contrastive loss to node representations, and DGI (Veličković et al., 2018), which learns graph-level embeddings by maximizing mutual information between node features and graph-level representations. However, these methods often rely on carefully designed augmentations and may require extra computational resources for generating and contrasting multiple views of each graph. Moreover, these methods typically rely on generating augmentations by modifying the graph, such as removing edges or nodes or perturbing node features. In the context of proteins, however, even a minor change in an amino acid can have a substantial impact on protein function. Thus, augmentations that disrupt the structure of the protein may lead to information loss. In contrast, our pretraining task does not require multiple views, augmentations, or masking strategies, thus simplifying the pretraining process and avoid the above limitations.

## 3 Methods

### 3.1 3D Graph Neural Networks

**Notation.** A 3D graph representing a protein is formally denoted as $G = (V, E, P)$, where $V$ represents the set of nodes, $E$ denotes the edges, and $P$ denotes the spatial coordinates of each node in the graph. In this work, we represent each amino acid as a node, using the position $\boldsymbol{p} \in \mathbb{R}^3$ of the $C_\alpha$ atom as the position of the amino acid. We Aconnect each node with the $k = 16$ nearest neighbors. We encode the amino acid

types as node features and the sequential distances as edge features. We denote as $\boldsymbol{h}_u^l$ the node features of node $u$ at layer $l$, and $\boldsymbol{e}_{uv}$ the edge feature vector for the edge $uv$. We denote as $N$ the total number of nodes and $\mathcal{N}_i$ the set of neighbors of node $i$. For a given node $v \in V$, the $k$-hop ego network of $v$ is the induced subgraph $G_v^{(k)} = (V_v^{(k)}, E_v^{(k)})$, where: $V_v^{(k)} = \{u \in V \mid \mathrm{dist}(v, u) \le k\}$ and $E_v^{(k)} = \{(u, w) \in E \mid u, w \in V_v^{(k)}\}$, where $\mathrm{dist}(v, u)$ denotes the shortest path distance between nodes $v$ and $u$ in $G$.

**Architecture.** We use two graph-based models that are specifically adapted for analyzing 3D protein structures, as the base models for our experiments. Specifically, we use ProNet (Wang et al., 2023), a recent 3D GNN model that achieves state-of-the-art performance in protein classification tasks. In each layer of ProNet, the node representations are updated as follows:

$$\boldsymbol{h}_i^{l+1} = f_1 \left( \boldsymbol{h}_i^l, \sum_{j \in \mathcal{N}_i} f_2 \left( \mathbf{v}_j^l, \mathbf{e}_{ji}, \mathcal{F} \left( d_{ji}, \theta_{ji}, \phi_{ji}, \tau_{ji} \right) \right) \right), \tag{1}$$

where $f_1$ and $f_2$ functions are parameterized using neural networks and $\mathcal{F}$ is a geometric transformation at the amino acid level. Here $(d_{ji}, \theta_{ji}, \phi_{ji})$ is the spherical coordinate of node $j$ in the local coordinate system of node $i$ to determine the relative position of $j$, and $\tau_{ji}$ is the rotation angle of edge $ji$.

The second base model is SchNet (Schütt et al., 2018), a popular invariant message passing GNN. SchNet performs message passing using element-wise multiplication of scalar features along with a radial filter that takes into account the pairwise distance $\|\vec{\boldsymbol{x}}_{ij}\|$ between two nodes. In each layer of SchNet, the node representations are updated as follows:

$$\boldsymbol{h}_i^{(l+1)} = \boldsymbol{h}_i^{(l)} + \sum_{j \in \mathcal{N}_i} f_1 \left( \boldsymbol{h}_j^{(l)}, \|\vec{\boldsymbol{x}}_{ij}\| \right) \tag{2}$$

Finally, we use also use a simple GCN model (Kipf & Welling, 2016), which updates the node representations as follows:

$$\mathbf{h}^{(l+1)} = f \left( \sum_{j \in \mathcal{N}(i) \cup \{i\}} \frac{1}{\sqrt{\hat{d}_j \hat{d}_i}} \mathbf{h}_j^{(l)} \right), \tag{3}$$

where $f$ is a linear projection followed by a non-linear activation and $\hat{d}_i = 1 + \sum_{j \in \mathcal{N}(i)} 1$.

The final protein representation, $\boldsymbol{h}_G$, for all models is computed by applying a sum pooling layer in the node representations from the last layer, $L$:

$$\boldsymbol{h}_G = \sum_{i=1}^{N} \boldsymbol{h}_i^L \tag{4}$$

## 3.2 Geometric Self-Supervised Pretraining

Pretraining plays a crucial role in enhancing the performance of deep neural networks, particularly in domains where labeled data is scarce or expensive to obtain. In this work, we leverage the large amount of available unlabeled 3D protein structures. Specifically, we pretrain the model to predict the distance between the geometric centroid of a subgraph and the geometric centroid of the entire protein $G$. The objective is to minimize the difference between the predicted and actual Euclidean distances.

In many real-world applications, ground truth distance measurements are subject to annotation noise, inherent measurement uncertainty, and prediction errors. Discretizing these continuous distances into bins allows our model to predict an interval rather than an exact numerical value, which in turn makes it less sensitive to minor deviations that could lead to large penalties in a regression framework. Additionally, using a cross-entropy loss for the classification task yields smoother gradients and more stable convergence during training than Mean Squared Error loss (Van Den Oord et al., 2016; Xiong & Yao, 2022). Therefore, we discretize the distances using 10 equal bins and formulate the problem as a classification task, using the cross-entropy loss instead. An overview of the proposed pipeline is illustrated in Figure 1.

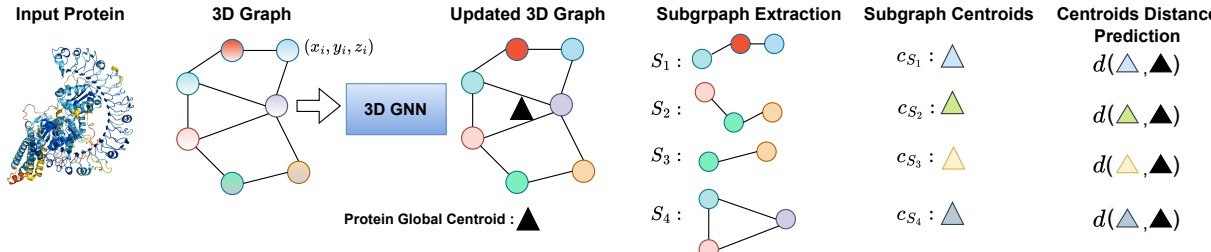

Figure 1: Visualization of the Geometric Centroid Pretraining Strategy for Protein Graph Neural Networks. This diagram illustrates the methodology employed to predict the Euclidean distances between the centroids of various subgraphs ($c_S$) and the overall protein centroid ($c_G$).

**Subgraph Computation.** While our approach is compatible with any subgraph selection method, for our implementation, we chose 2-hop ego networks centered around 10% of the amino acids in each protein. Therefore, for each protein $G$, we obtain a set of different subgraphs $\mathcal{S}_\mathcal{G}$, where each subgraph corresponds to a 2-hop ego network.

Firstly, we compute the geometric centroid of the protein and the subgraphs. The geometric centroid $c_G$ of the protein is calculated by averaging the coordinates of all amino acids in the protein:

$$\begin{aligned}
\boldsymbol{c}_G &= \frac{1}{|V|} \sum_{i \in V} \boldsymbol{p}_i \\
\boldsymbol{c}_G &= \left( \frac{1}{|V|} \sum_{i \in V} x_i, \frac{1}{|V|} \sum_{i \in V} y_i, \frac{1}{|V|} \sum_{i \in V} z_i \right)
\end{aligned} \tag{5}$$

where $(x_i, y_i, z_i)$ are the coordinates of each node $i$. Similarly, the centroid $c_S$ for each subgraph $S \in \mathcal{S}_\mathcal{G}$ is calculated by averaging the coordinates of the nodes within the subgraph:

$$\boldsymbol{c}_S = \left( \frac{1}{|S|} \sum_{j \in S} x_j, \frac{1}{|S|} \sum_{j \in S} y_j, \frac{1}{|S|} \sum_{j \in S} z_j \right), \tag{6}$$

where $|S|$ is the number of nodes in subgraph $S$. We then compute the Euclidean distance between the centroid of the protein and the centroid of each subgraph, $S$:

$$d(\mathbf{c}_S, \mathbf{c}_G) = \|\mathbf{c}_S - \mathbf{c}_G\| \tag{7}$$

Then the label for each subgraph is computed by discretizing this distance into one of 10 equal bins, which transforms the regression task into a classification task.

**Distance Prediction.** To predict the distances, we calculate the embedding for a subgraph $S$ by aggregating the node representations within this subgraph:

$$\boldsymbol{h}_S = \sum_{i \in S} \boldsymbol{h}_i^L \tag{8}$$

This summation operation merges the features of the nodes in the subgraph from the final layer $L$ of ProNet to a vector that represents the entire subgraph. The predicted probability for each bin is derived from the embeddings $\mathbf{h}_G$ and $\mathbf{h}_S$, using a parameterized function $f(\mathbf{h}_S \| \mathbf{h}_G)$. In our experiments, we use a two-layer multilayer perceptron (MLP) to parameterize the function $f$. The loss function is defined as the cross-entropy loss between the true and predicted bin labels across all proteins and their respective subgraphs:

$$\mathcal{L}_{\text{pretraining}} = -\frac{1}{N} \sum_{G \in \mathcal{D}} \sum_{S \in \mathcal{S}_G} \sum_{k=1}^{10} y_{S,G}^{(k)} \log \hat{y}_{S,G}^{(k)}, \tag{9}$$

where $\mathcal{D}$ is the collection of training protein graphs, $N$ is the number of subgraphs, $y_{S,G}^{(k)}$ is the true probability for bin $k$ (1 for the correct bin, 0 otherwise), and $\hat{y}_{S,G}^{(k)}$ is the predicted probability for bin $k$.

**Complexity.** The additional overhead introduced by our method due to the subgraph computation can be eliminated by performing it once, as a preprocessing step, by storing the subgraphs. Moreover, since we extract the subgraph representations from the final node representations of the GNN, we only require one forward pass for each graph.

**Motivation.** In this work, we aim to address the limitations inherent in traditional pretraining methods for protein representation learning. Existing approaches often rely on simplistic masking strategies that can not accurately capture the complex three-dimensional structural patterns in proteins. These methods tend to overlook the spatial relationships and the hierarchical organization within protein structures, as they focus solely on single node or edge masking. Moreover, contrastive learning methods rely on generating multiple augmented views of the same graph by altering its structure, such as by removing nodes or edges or by perturbing node features, assuming that small edits preserve the semantic meaning. While these augmentations can be effective in other domains, in proteins, even minor modifications can dramatically affect function, risking the loss of critical biological information. In contrast, our pretraining approach avoids such potentially disruptive augmentations by directly leveraging the inherent geometric and hierarchical relationships within the protein structure.

## 4 Experiments and results

### 4.1 Datasets

**Pretraining Dataset** For the pretraining, we used 542k SwissProt proteins structures from the AlphaFold Database (Varadi et al., 2022). This dataset offers high-quality, predicted protein structures, making it a reliable choice for model training. The pretraining process captures a broad spectrum of structural and functional patterns, which is crucial for the generalization to other proteins.

**Fold Classification.** We used the dataset and experimental protocols from (Wang et al., 2023). The dataset encompasses a total of 16,712 proteins categorized into 1,195 different folds. Our evaluation spans three distinct test sets: Fold, Superfamily, and Family. For the Fold Dataset, we used the same dataset as in previous studies (Hermosilla et al., 2020; Wang et al., 2023). To assess the models ability to generalize, three test sets are used: Fold, where proteins from the same superfamily are not seen during training; Superfamily, where proteins from the same family are excluded from training; and Family, where proteins from the same family are included in the training data. Among these, the Fold test set presents the highest challenge due to its significant divergence from the training set's conditions. For this task, the dataset is divided into 12,312 proteins for training, 736 for validation, and additional subsets for testing: 718 proteins for the Fold test, 1,254 for Superfamily, and 1,272 for Family.

**React Classification.** An Enzyme Commission (EC) number is a numerical classification scheme for enzymes, based on the chemical reactions they catalyze. Each protein in the dataset is associated with an EC number, with annotations for these numbers obtained from the SIFTS database (Dana et al., 2019). The dataset encompasses a total of 37,428 proteins representing 384 distinct EC numbers. We utilized a dataset comprised of 3D protein structures sourced from the Protein Data Bank (PDB) (Berman et al., 2000). Following the experimental setup of (Wang et al., 2023), 29,215 proteins were used for training, 2,562 for validation, and 5,651 for testing. Every EC number is represented across all three dataset splits. Proteins with more than 50% similarity were grouped together in the same split. This setup aids in evaluating the model's ability to generalize across different protein structures.

### 4.2 Experimental Setup

**Baselines.** We compare our pretraining method with the edge distance pretraining task and with In-foGraph(Sun et al., 2019) contrastive pertaining approach. Edge distance prediction is a self-supervised learning task in graph representation learning, aimed at predicting the pairwise distance between two nodes

in a graph. In this task, we sample 256 edges from each batch, a mask is applied on the sampled edges(and their associated attributes), and the distance is then predicted based on the learned node representations of these sampled edges. Both subgraph distance prediction and edge distance prediction aim to learn geometric or distance information, so we chose edge distance prediction as a relevant comparison for evaluating the effectiveness of our approach. InfoGraph is a popular graph contrastive learning method that aims to maximize mutual information between local substructures and the global graph representation to learn informative graph embeddings. We pretrain all models on the same pertaining dataset from the AlphaFold database.

We use ProteinWorkshop library to run all the experiments, including model pretraining and downstream classification tasks. ProteinWorkshop provides various protein representation learning benchmarks, with implementation of numerous featurisation schemes, datasets and tasks. We use ProNet, SchNet and GCN as the base architectures. We further implement the ProNet model and our self-supervised pretraining task in the ProteinWorkshop library to have a fair comparison. We choose three $C\alpha$-based featurisation schemes: ca_base uses one-hot encoding of the amino acid type for each node; ca_angles added 16-dimensional positional encoding and $\kappa, \alpha \in \mathbb{R}^4$ the virtual torsion and bond angles defined over $C\alpha$ atoms; ca_bb added $\phi, \psi, \omega \in \mathbb{R}^6$ which correspond to the backbone dihedral angles.

**Training Details.** For ProNet, we use the best hyperparameters from (Wang et al., 2023) and apply only ca_base featurisation as it computes internally angle information. For GCN and SchNet, we applied all featurisation methods and used the default hyperparameters from ProteinWorkshop. For all pretraining tasks, we conducted a grid search to determine the optimal learning rate from $1e-4$ and $3e-4$. For the edge distance task, we select 256 edges to be masked from the batch. For all tasks, pretraining is performed for 10 epochs with batch size 32 using a linear warm-up with a cosine schedule. For downstream tasks, we search for every model and featurisation the best learning rate among $0.00001, 0.0001, 0.0003, 0.001$ and the best dropout among $0.0, 0.1, 0.3, 0.5$ based on validation performance on the fold classification task, we use 150 maximum number of epochs with a batch size of 32 and ReduceLROnPlateau learning rate scheduler monitoring the validation metric with patience of 5 epochs and reduction of 0.6. We monitor the validation accuracy and perform early stopping with patience of 10 epochs, we report the average and standard deviation over three runs using different seeds.

Table 1: Accuracy (%) and F1_max (%) on reaction and fold classification tasks with **ca_base featurization**.

| Model | Pretraining | React | | Fold | | | | | |
| --- | --- | --- | --- | --- | --- | --- | --- | --- | --- |
| | | Accuracy | F1_max | Fold | | Super-Family | | Family | |
| | | | | Accuracy | F1_max | Accuracy | F1_max | Accuracy | F1_max |
| GCN | None | 43.44±2.1 | 50.57±2.33 | 12.32±0.6 | 16.99±0.73 | 10.85±0.1 | 16.84±0.17 | 57.35±1.8 | 64.55±1.54 |
| | Edge Distance | 43.39±1.3 | 51.89±2.05 | 12.49±0.2 | 17.47±0.40 | 11.39±0.5 | 16.47±0.32 | 54.88±4.9 | 61.85±5.01 |
| | InfoGraph | 43.70±0.9 | 53.70±0.9 | 11.17±0.4 | 16.31±1.0 | 11.32±0.3 | 16.50±0.9 | 57.83±4.9 | 63.36±4.6 |
| | Subgraph Distance (Ours) | **47.46±0.9** | **54.47±0.83** | **12.90±0.1** | **17.49±0.66** | **11.81±0.5** | **17.23±0.07** | **58.40±4.2** | **66.90±1.79** |
| ProNet | None | 77.96±5.3 | 78.10±1.9 | 46.92±1.4 | 47.38±2.53 | 60.32±0.1 | 58.30±1.61 | 97.69±0.1 | 96.62±0.63 |
| | Edge Distance | 79.14±2.3 | 79.89±2.5 | 47.40±1.1 | 47.24±3.57 | 63.13±1.1 | 57.20±0.98 | **98.07±0.1** | 95.72±0.33 |
| | InfoGraph | 75.50±1.0 | 76.88±2.1 | 39.39±1.5 | 47.60±1.2 | 52.30±0.9 | 59.65±1.3 | 95.39±0.1 | 97.25±0.3 |
| | Subgraph Distance (Ours) | **80.61±1.3** | **81.10±1.4** | **50.11±1.0** | **49.38±0.39** | **64.79±2.7** | **61.76±1.99** | 97.88±0.1 | **98.08±0.25** |
| SchNet | None | 59.48±1.9 | 66.04±1.63 | 21.35±2.3 | 27.43±1.19 | 23.53±0.3 | 29.76±0.43 | 76.85±1.7 | 83.35±1.22 |
| | Edge Distance | 60.95±1.9 | 67.67±1.50 | 22.16±1.5 | **30.16±0.77** | **29.36±1.7** | **35.19±0.46** | 79.60±1.3 | 84.10±1.43 |
| | InfoGraph | 64.47±2.2 | **70.60±2.2** | 23.20±0.6 | 29.33±0.7 | 28.64±0.3 | 34.79±0.3 | 81.96±1.5 | **86.49±1.1** |
| | Subgraph Distance (Ours) | **65.03±1.3** | 68.73±1.91 | **23.41±0.2** | 29.27±1.31 | 27.65±1.0 | 32.94±0.28 | **82.62±1.7** | 83.99±0.34 |

## 4.3 Results and Discussion

**Downstream Task Results.** We report the accuracy and F1 max results for different featurizaton schemes in Tables 1, 2 and 3. Compared to models without pretraining and those using edge distance pretraining, the subgraph distance method consistently yields higher performance. Specifically, SchNet pretrained with our task can lead to significant improvements in accuracy such as 4.78% in the Super-Family task with ca_angles featurization and 5.85% with ca_bb featurization. The same patterns hold for GCN and ProNet, where our pretrained models are significantly better, demonstrating that the hierarchical and geometric information

Table 2: Accuracy(%) and F1_max on reaction and fold classification tasks with **ca_angles featurization**.

| Model | Pretraining | React | | Fold | | | | | |
|---|---|---|---|---|---|---|---|---|---|
| | | Accuracy | F1_max | Fold | | Super-Family | | Family | |
| | | | | Accuracy | F1_max | Accuracy | F1_max | Accuracy | F1_max |
| GCN | None | 70.14±1.3 | 75.81±1.37 | 25.45±0.7 | 31.28±0.55 | 33.21±1.3 | 40.63±1.19 | 89.68±0.5 | 93.06±0.56 |
| | Edge Distance | 69.40±1.0 | 75.86±0.04 | 24.73±0.5 | 30.72±0.53 | 33.84±1.3 | 40.82±0.80 | 88.71±0.7 | 92.23±0.20 |
| | InfoGraph | **75.55±1.4** | **80.31±1.1** | 27.24±1.2 | 33.16±0.5 | **37.91±0.4** | **45.07±0.4** | 90.86±0.4 | 93.12±0.1 |
| | Subgraph Distance (Ours) | 70.75±1.3 | 76.71±1.69 | **27.67±0.5** | **33.29±0.62** | 35.99±0.7 | 43.24±1.02 | **91.07±0.2** | **93.67±0.40** |
| SchNet | None | 69.27±3.1 | 75.06±2.56 | 26.66±0.8 | 33.48±0.95 | 34.87±0.8 | 41.97±0.40 | 90.29±0.7 | 93.22±0.61 |
| | Edge Distance | 68.81±2.8 | 75.33±0.72 | 27.89±0.4 | 34.41±0.54 | 36.19±0.9 | 43.18±1.39 | 90.21±0.3 | 92.95±0.35 |
| | InfoGraph | 72.25±1.8 | 77.13±1.2 | 30.10±2.0 | 36.42±0.8 | 39.42±0.1 | **46.59±0.1** | 91.61±0.3 | **94.63±1.2** |
| | Subgraph Distance (Ours) | **72.26±2.3** | **77.50±2.11** | **31.22±1.9** | **37.04±1.44** | **39.65±0.3** | 46.44±0.67 | **91.94±0.0** | 94.45±0.19 |

Table 3: Accuracy(%) and F1_max on reaction and fold classification tasks with **ca_bb featurization**.

| Model | Pretraining | React | | Fold | | | | | |
|---|---|---|---|---|---|---|---|---|---|
| | | Accuracy | F1_max | Fold | | Super-Family | | Family | |
| | | | | Accuracy | F1_max | Accuracy | F1_max | Accuracy | F1_max |
| GCN | None | 70.82±0.9 | 76.56±1.06 | 26.18±1.4 | 32.21±0.78 | 33.43±0.2 | 40.21±0.25 | 89.90±0.5 | 92.79±0.28 |
| | Edge Distance | 69.80±3.8 | 77.85±0.65 | 24.21±1.0 | 31.35±0.46 | 32.31±1.5 | 40.04±0.58 | 88.31±1.8 | 91.55±0.54 |
| | InfoGraph | **74.18±1.1** | **79.13±0.7** | 28.27±0.2 | 32.88±0.5 | 35.42±0.0 | **44.86±0.4** | **92.21±0.8** | **94.51±0.5** |
| | Subgraph Distance (Ours) | 71.44±0.5 | 77.26±0.31 | 27.69±0.3 | **33.54±0.11** | 35.77±0.7 | 42.74±0.34 | 90.92±0.9 | 93.34±0.48 |
| SchNet | None | 70.33±0.5 | 76.40±2.67 | 28.43±0.6 | 33.84±0.75 | 36.28±0.3 | 42.51±0.88 | 89.94±0.8 | 92.36±0.22 |
| | Edge Distance | 73.72±0.8 | 78.66±0.99 | 31.46±1.3 | 37.60±1.49 | 38.93±1.5 | 45.78±1.39 | 90.12±1.3 | 93.00±0.57 |
| | InfoGraph | 70.89±0.3 | 78.03±1.4 | **31.90±0.2** | **38.49±0.9** | 38.77±1.2 | 45.49±0.2 | 89.79±0.4 | 92.83±0.8 |
| | Subgraph Distance (Ours) | **73.78±0.5** | **78.76±2.05** | 31.45±1.0 | 37.02±0.64 | **42.13±1.6** | **47.59±0.05** | **91.99±0.5** | **94.79±0.29** |

captured through subgraph distance pretraining is beneficial. We further outperform InfoGraph in most cases, while our approach is around 30 times faster. We further provide results with additional evaluation metrics in the Appendix A.

**Pretraining Analysis.** In this section, we analyze the performance of our model in the pretraining task. Specifically, we present accuracy and loss curves across the pretraining epochs for the test set, and provide a confusion matrix to further understand the quality of predictions.

We pretrained the SchNet model with the three different feature schemes and we plot the accuracy and cross-entropy loss in Figure 2. The loss curves demonstrate that for all feature schemes the loss is decreasing during pretraining, with schnet_ca_bb showing the lowest final loss. In Figure 3, we report the results for the GCN model during pretraining. Similarly, we observe that the loss is decreasing and the accuracy is increasing during pretraining. The ca_angles and ca_bb schemes achieve the best performance, which is reasonable as they have more information for the geometry of the protein.

Interestingly, we observe that there is a correlation between the pretraining performance and the downstream tasks performance. Models pretrained with the ca_angles and ca_bb featurization schemes, which show higher pretraining accuracies, generally exhibit better performance on downstream tasks. This result aligns with observations in language modeling, where strong pretraining performance, such as accurate next-word prediction, is a reliable indicator of the performance in various downstream tasks (Wei et al., 2021). Our study extends this concept geometric self-supervised learning, demonstrating that accurate prediction of subgraph distances during pretraining can significantly enhance the performance of models in downstream applications.

## 4.4 Ablation Studies

**Impact of Different Subgraph Extraction Methods.** Our proposed approach relies on extracting 2-hop ego networks as subgraphs to capture both local and contextual structural information. To evaluate the importance of this choice, we compare it against two alternative strategies, using *1-hop ego networks* and *random subgraphs*. In the 1-hop ego network method, we extract only the immediate neighbors of the selected central nodes. This results in smaller subgraphs that capture only local structural information. In

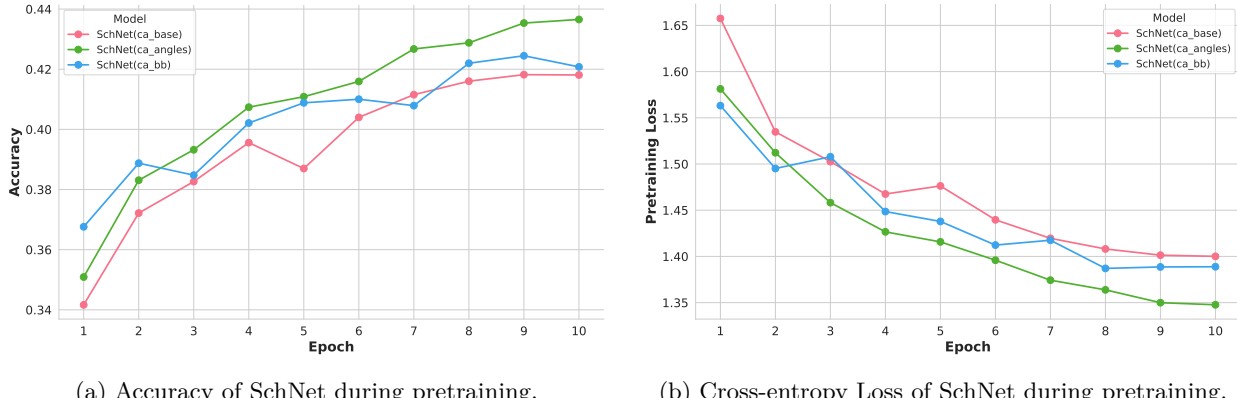

(a) Accuracy of SchNet during pretraining.      (b) Cross-entropy Loss of SchNet during pretraining.

Figure 2: Illustration of the key performance metrics of the SchNet Models with different featurization schemes, during pretraining.

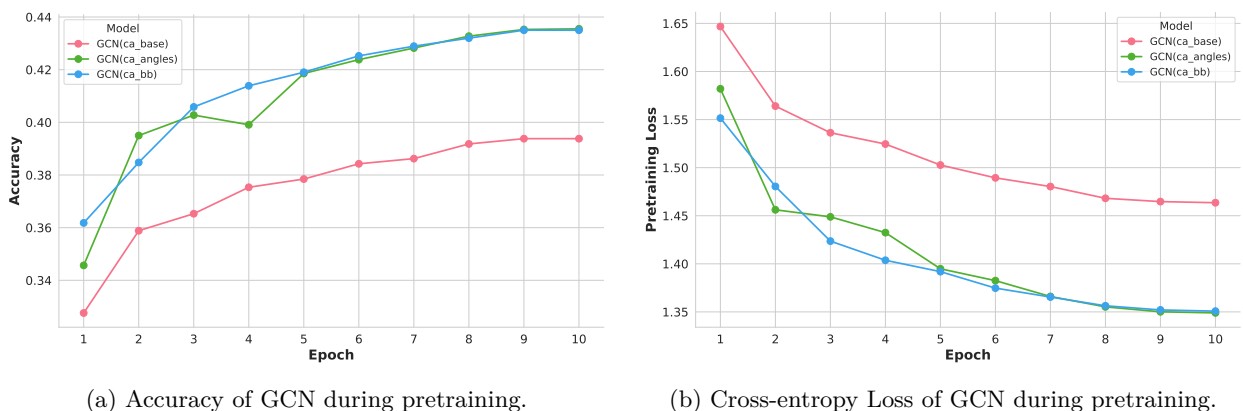

(a) Accuracy of GCN during pretraining.      (b) Cross-entropy Loss of GCN during pretraining.

Figure 3: Illustration of the key performance metrics of the GCN Models with different featurization schemes, during pretraining.

the random subgraph method, we randomly sample 20 nodes from the protein graph without considering connectivity. The geometric centroid of this random subgraph is then computed in the same way as in our original method. These comparisons allow us to systematically evaluate how different subgraph selection methods influence pretraining effectiveness and downstream task performance. We present the results in Table 4. Using 2-hop ego networks consistently leads to better performance in the downstream tasks. The 1-hop ego networks, as they are too small, they fail to encode meaningful structural and geometric relationships, leading to weaker downstream performance. Meanwhile, random subgraphs may not capture biologically relevant structural motifs, further degrading performance.

**Classification vs. Regression for Distance Prediction.** In our primary experiments, we formulated the distance prediction task as a classification problem by discretizing the distance values into 10 bins. An alternative approach would be to treat it as a regression task, directly predicting the continuous Euclidean distance between subgraph centroids and the protein centroid. To assess the impact of this choice, we compare the classification-based approach against a regression-based alternative. Table 5 presents the results of this experiment. Our findings demonstrate that the classification formulation consistently outperforms regression.

**Impact of Centroid Computation Methods.** To evaluate the impact of different centroid computation strategies on our pretraining framework, we conducted an ablation study comparing several methods for computing the protein and subgraph centroids. We present the results in Table 6.

Table 4: Ablation Study on the effect of different subgraph extraction methods for the proposed pertaining strategy.

| Model | Subgraph Extraction Method | React | Fold | | |
|---|---|---|---|---|---|
| | | | Fold | Super-Family | Family |
| GCN Pretrained | 2-hop Ego Networks | 47.46± 0.90 | **12.90± 0.10** | **11.81± 0.50** | 58.40± 4.20 |
| | 1-hop Ego Networks | **49.86± 0.76** | 11.95± 1.74 | 11.35± 0.35 | **61.03± 2.31** |
| | Random Subgraphs | 44.60± 0.99 | 12.32± 0.95 | 11.49± 0.32 | 59.87± 3.10 |
| ProNet Pretrained | 2-hop Ego Networks | **80.61±1.30** | **50.11±1.00** | **64.79±2.70** | **97.88± 0.10** |
| | 1-hop Ego Networks | 75.20±1.10 | 42.27±1.37 | 56.45±0.65 | 96.29±0.38 |
| | Random Subgraphs | 74.11±0.80 | 38.22±1.44 | 50.34±2.04 | 94.37±0.33 |
| SchNet Pretrained | 2-hop Ego Networks | **65.03± 1.30** | **23.41± 0.20** | **27.65± 1.00** | **82.62±1.70** |
| | 1-hop Ego Networks | 61.61± 2.62 | 22.05± 1.21 | 26.56± 1.31 | 80.78± 0.98 |
| | Random Subgraphs | 57.77± 2.71 | 23.30± 1.12 | 26.39± 0.49 | 79.10± 0.83 |

Table 5: Ablation Study on the effect of Classification vs. Regression pretraining objectives in the proposed pertaining strategy.

| Model | Pretraining Objective | React | Fold | | |
|---|---|---|---|---|---|
| | | | Fold | Super-Family | Family |
| GCN Pretrained | Regression | 45.54±3.2 | 12.12±1.8 | 11.32±0.8 | 58.22±1.1 |
| | Classification | **47.46±0.9** | **12.90±0.1** | **11.81±0.5** | **58.40±4.2** |
| ProNet Pretrained | Regression | 78.76±1.2 | 47.80±0.01 | 62.45±2.05 | 98.19±0.14 |
| | Classification | **80.61±1.3** | **50.11±1.0** | **64.79±2.7** | **97.88±0.1** |
| SchNet Pretrained | Regression | 55.31±1.64 | 22.07±1.78 | 24.95±0.37 | 79.93±0.11 |
| | Classification | **65.03±1.3** | **23.41±0.2** | **27.65±1.0** | **82.62±1.7** |

Table 6: Accuracy on reaction and fold classification tasks with **ca_base featurization** using different method to compute the centroids of the graph.

| Model | Centroid Method | React | Fold | | |
|---|---|---|---|---|---|
| | | | Fold | Super-Family | Family |
| GCN Pretrained | Arithmetic Mean | **47.46±0.9** | **12.90±0.1** | 11.81±0.5 | 58.40±4.2 |
| | Geometric Mean | 45.49±1.6 | 12.36±0.8 | 11.75±0.4 | 59.12±4.3 |
| | Median | 46.38±1.2 | 12.57±0.3 | **12.39±1.1** | **61.79±1.8** |
| ProNet Pretrained | Arithmetic Mean | **80.61±1.3** | **50.11±1.0** | **64.79±2.7** | **97.88±0.1** |
| | Geometric Mean | 74.33±0.9 | 48.63 ±0.8 | 64.63±2.1 | 97.78±0.3 |
| | Median | 74.56±1.2 | 46.78±1.1 | 60.52±1.8 | 95.52±0.3 |
| SchNet Pretrained | Arithmetic Mean | **65.03±1.3** | 23.41±0.2 | **27.65±1.0** | **82.62±1.7** |
| | Geometric Mean | 58.26±0.8 | **23.87±0.7** | 25.81±0.3 | 80.73±1.2 |
| | Median | 59.59±1.4 | 22.04±0.7 | 26.50±0.7 | 80.93±0.9 |

## 5 Conclusion and Future work

In this work, we proposed a new self-supervised learning method to learn accurate protein representations from 3D structures. By capitalizing on the extensive collection of protein structures available, we pre-trained a 3D GNN model to predict the distance between the geometric centroid of the entire protein and various

subgraphs within the protein. We experimentally show that our pretraining strategy leads to improved performance in downstream classification tasks, such as protein fold and reaction classification, while also outperforming typical pretraining methods such as edge masking. In future work, we plan to explore the effects of various subgraph selection strategies and investigate how combining our approach with additional pretraining tasks could further enhance performance. We hope that our work will inspire more people to leverage the large amount of protein structures and develop specialized self-supervised learning methods for these data.

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

# A  Appendix

## A.1  Additional Metrics

Table 7: Macro F1 score on reaction and fold classification tasks with **ca_base featurization**.

| Model | Pretraining | React | Fold | | |
|---|---|---|---|---|---|
| | | | **Fold** | **Super-Family** | **Family** |
| GCN | None | 27.61±1.87 | 2.78±0.25 | 2.74±0.13 | 24.36±1.54 |
| | Edge Distance | 28.81±1.71 | **3.07±0.26** | 2.90±0.33 | 22.59±2.61 |
| | Subgraph Distance (Ours) | **31.15±0.78** | 3.05±0.14 | **3.14±0.14** | **25.79±0.96** |
| ProNet | None | 47.07±1.33 | 13.28±1.61 | 20.78±1.54 | 77.19±4.19 |
| | Edge Distance | 48.02±1.23 | 14.25±1.06 | 20.36±0.49 | 74.48±1.01 |
| | Subgraph Distance (Ours) | **48.56±1.38** | **14.85±0.89** | **22.31±2.84** | **83.46±0.81** |
| SchNet | None | 42.27±1.61 | 5.53±0.78 | 6.73±0.27 | 44.47±2.62 |
| | Edge Distance | 43.87±1.80 | **6.90±0.33** | **9.06±0.46** | **45.64±3.18** |
| | Subgraph Distance (Ours) | **44.42±1.92** | 6.65±0.64 | 7.57±0.29 | 42.52±0.62 |

Table 8: Macro F1 score on reaction and fold classification tasks with **ca_angles featurization**.

| Model | Pretraining | React | Fold | | |
|---|---|---|---|---|---|
| | | | **Fold** | **Super-Family** | **Family** |
| GCN | None | 54.93±1.21 | 7.19±0.65 | 10.76±0.63 | 60.94±2.35 |
| | Edge Distance | 54.16±0.87 | 7.11±0.41 | 11.12±0.62 | 57.89±1.36 |
| | Subgraph Distance (Ours) | **55.68±1.34** | **7.62±0.41** | **12.42±0.43** | **63.37±0.94** |
| SchNet | None | 53.40±3.07 | 7.17±0.35 | 11.46±0.37 | 62.15±2.26 |
| | Edge Distance | 53.61±1.98 | 7.70±0.39 | 11.78±0.73 | 61.23±1.59 |
| | Subgraph Distance (Ours) | **56.35±2.87** | **8.75±1.38** | **13.86±0.54** | **65.73±1.27** |

Table 9: Macro F1 score on reaction and fold classification tasks with **ca_bb featurization**.

| Model | Pretraining | React | Fold | | |
|---|---|---|---|---|---|
| | | | **Fold** | **Super-Family** | **Family** |
| GCN | None | 55.55±1.21 | 7.27±0.47 | 10.96±0.25 | 61.55±0.86 |
| | Edge Distance | **56.95±0.54** | 6.99±0.36 | 10.61±0.35 | 58.30±2.85 |
| | Subgraph Distance (Ours) | 56.57±0.84 | **7.83±0.15** | **12.15±0.50** | **64.25±1.96** |
| SchNet | None | 55.10±3.66 | 7.78±0.28 | 11.75±0.43 | 60.89±2.56 |
| | Edge Distance | **58.45±0.94** | **8.62±0.71** | 13.23±0.53 | 62.07±3.09 |
| | Subgraph Distance (Ours) | 57.53±2.19 | 8.77±0.57 | **13.81±0.29** | **66.69±1.42** |

Table 10: Weighted ROC AUC on reaction and fold classification tasks with **ca_base featurization**.

| Model | Pretraining | React | Fold | | |
| --- | --- | --- | --- | --- | --- |
| | | | **Fold** | **Super-Family** | **Family** |
| GCN | None | 94.48±0.33 | 67.86±0.24 | **75.97±0.09** | 78.17±0.57 |
| | Edge Distance | 94.50±0.59 | 67.69±0.20 | 74.74±0.07 | 77.19±1.55 |
| | Subgraph Distance (Ours) | **95.27±0.20** | **68.83±0.39** | 75.85±1.60 | **78.96±0.42** |
| ProNet | None | 97.81±0.57 | 91.07±0.53 | 93.80±0.38 | 82.40±0.03 |
| | Edge Distance | 97.92±0.43 | 89.78±1.53 | 93.29±0.62 | 82.29±0.06 |
| | Subgraph Distance (Ours) | **98.01±0.23** | **91.66±0.47** | **95.02±0.42** | **82.43±0.02** |
| SchNet | None | 96.73±0.04 | 75.56±0.68 | 82.43±0.37 | 80.83±0.20 |
| | Edge Distance | 97.13±0.06 | **78.94±0.18** | **85.12±0.63** | **81.22±0.03** |
| | Subgraph Distance (Ours) | **97.25±0.24** | 78.36±0.93 | 84.09±0.49 | 81.20±0.06 |

Table 11: Weighted ROC AUC on reaction and fold classification tasks with **ca_angles featurization**.

| Model | Pretraining | React | Fold | | |
| --- | --- | --- | --- | --- | --- |
| | | | **Fold** | **Super-Family** | **Family** |
| GCN | None | 97.49±0.08 | 79.85±0.51 | 86.92±0.36 | 81.93±0.05 |
| | Edge Distance | 97.41±0.28 | 80.36±0.27 | 86.22±0.08 | 81.78±0.13 |
| | Subgraph Distance (Ours) | **97.62±0.16** | **81.45±0.31** | **88.06±0.36** | **85.38±5.73** |
| SchNet | None | 97.67±0.02 | 82.29±0.70 | 88.75±0.37 | 82.14±0.12 |
| | Edge Distance | 97.57±0.06 | 83.38±0.31 | 89.19±0.55 | 81.19±0.08 |
| | Subgraph Distance (Ours) | **97.73±0.21** | **84.12±0.50** | **90.50±0.17** | **82.26±0.00** |

Table 12: Weighted ROC AUC reaction and fold classification tasks with **ca_bb featurization**.

| Model | Pretraining | React | Fold | | |
| --- | --- | --- | --- | --- | --- |
| | | | **Fold** | **Super-Family** | **Family** |
| GCN | None | 97.37±0.06 | 80.89±0.37 | 86.59±0.10 | 81.85±0.05 |
| | Edge Distance | 97.44±0.05 | 80.70±0.44 | 86.35±0.22 | 81.67±0.01 |
| | Subgraph Distance (Ours) | **97.50±0.37** | **82.06±0.29** | **87.95±0.13** | **82.02±0.06** |
| SchNet | None | 97.63±0.85 | 82.66±0.14 | 89.12±0.19 | 82.01±0.06 |
| | Edge Distance | 97.77±0.18 | **85.41±0.68** | 90.40±0.26 | 82.04±0.37 |
| | Subgraph Distance (Ours) | **97.84±0.07** | 85.13±1.06 | **90.81±0.41** | **82.26±0.05** |

## A.2 Ablation Study: Edge Masking Baseline.

We evaluated the impact of the number of masked edges of the edge distance baseline by systematically varying the proportion of masked edges during pretraining. In particular, we considered three masking configurations: masking 10% of the edges, masking 20% of the edges, and using a fixed value of 256 masked edges per batch. Tables 13, 14, 15 summarizes the results. Our results indicate that the performance of the edge distance baseline is not highly sensitive to the masking rate.

Table 13: Accuracy (%) on Reaction and Fold Classification Tasks with **ca_base featurization** under different number of masked edges for the Edge Distance Baseline.

| Model | Masked Edges | React | Fold | Super-Family | Family |
|-------|--------------|-------|------|--------------|--------|
| GCN (Edge Distance) | 10% | 39.74 | 11.64 | 10.01 | 53.38 |
| | 20% | 38.35 | 11.37 | 10.72 | 50.32 |
| | 256 Fixed | **43.39** | **12.49** | **11.39** | **54.88** |
| ProNet (Edge Distance) | 10% | 79.58 | 47.68 | **63.91** | 96.00 |
| | 20% | **79.98** | **48.12** | 63.31 | 96.19 |
| | 256 Fixed | 79.14 | 47.40 | 63.13 | **98.07** |
| SchNet (Edge Distance) | 10% | 63.48 | 23.21 | **30.02** | **82.47** |
| | 20% | **64.19** | 22.41 | 27.26 | 81.14 |
| | 256 Fixed | 60.95 | 22.16 | 29.36 | 79.60 |

Table 14: Accuracy (%) on Reaction and Fold Classification Tasks with **ca_angles featurization** under different number of masked edges for the Edge Distance Baseline.

| Model | Masked Edges | React | Fold | Super-Family | Family |
|-------|--------------|-------|------|--------------|--------|
| GCN (Edge Distance) | 10% | 67.51 | 23.30 | 28.10 | 82.41 |
| | 20% | 66.40 | 24.14 | 31.87 | 85.72 |
| | 256 Fixed | **69.40** | **24.73** | **33.84** | **88.71** |
| SchNet (Edge Distance) | 10% | 67.77 | 27.22 | 37.85 | **91.34** |
| | 20% | 68.73 | **28.26** | **39.58** | 89.89 |
| | 256 Fixed | **68.81** | 27.89 | 36.19 | 90.21 |

Table 15: Accuracy (%) on Reaction and Fold Classification Tasks with **ca_bb featurization** under different number of masked edges for the Edge Distance Baseline.

| Model | Masked Edges | React | Fold | Super-Family | Family |
|-------|--------------|-------|------|--------------|--------|
| GCN (Edge Distance) | 10% | 66.98 | **24.28** | **30.62** | 86.19 |
| | 20% | 68.97 | 23.13 | 30.15 | 86.82 |
| | 256 Fixed | **69.80** | 24.21 | 32.31 | **88.31** |
| SchNet (Edge Distance) | 10% | 73.05 | 29.79 | 35.45 | **90.23** |
| | 20% | 69.34 | 23.13 | 30.15 | 86.82 |
| | 256 Fixed | **73.72** | **31.46** | **38.93** | 90.12 |

