# OpenReview forum: "Geometric Self-Supervised Pretraining on 3D Protein Structures using Subgraphs"
_TMLR — Rejected by TMLR_

### Review · Reviewer_46M4 · 2024-12-22

**Summary Of Contributions:**

The paper introduces a novel geometric self-supervised pretraining method for learning protein representations using 3D protein structures. The approach predicts distances between the geometric centroids of protein subgraphs and the global centroid of the protein to pretrain 3D graph neural networks (GNNs). This method emphasizes the hierarchical and geometric properties of proteins, improving performance in downstream protein classification tasks without relying on masking strategies or augmentations.

**Audience:**

Yes

**Claims And Evidence:**

No

**Requested Changes:**

1. Please explain the reason behind selecting 2-hop ego networks for subgraph construction. Are there specific advantages of this approach compared to other subgraph selection strategies?
2. Please provide an explanation for the decision to treat all amino acids equally in the model. How does this simplification impact the model’s ability to capture the unique biochemical properties of individual amino acids?
3. If possible, please compare the proposed pretraining method to other state-of-the-art pretraining techniques, such as contrastive learning or hybrid strategies.

**Strengths And Weaknesses:**

**Strengths**
1. The paper proposes a new pretraining task that leverages geometric distances between subgraphs and the global protein structure.
2. The pretraining strategy is compatible with multiple GNN architectures (ProNet, SchNet, and GCN), showcasing its adaptability.
3. The method outperforms baseline approaches, with significant improvements (up to 6%) in various protein classification benchmarks.
4. The paper is well-written.

**Weaknesses**
1. The paper primarily uses 2-hop ego networks as subgraphs for the pretraining task but does not deeply explore or justify why this specific choice is optimal. Other subgraph selection strategies might yield better performance or provide insights into specific protein features.
2. When computing the centroids, the method assumes a uniform treatment of all amino acids, focusing on their geometric centroids without incorporating their biochemical properties or functional roles. This may overlook critical distinctions between amino acids that might influence protein folding, interactions, and functionality. Such a simplification might limit the model’s ability to capture biologically relevant patterns.
3. While the proposed subgraph distance pretraining is compared to edge distance prediction, the experiments do not include comparisons with other pretraining strategies, such as contrastive learning or multi-task pretraining. This makes it hard to determine whether the proposed method outperforms the existing state-of-the-art pre-training methods.

---

> ### Author Response · Authors · 2025-03-05
> **Response 1/n**
>
> We would like to sincerely thank the reviewer for the thoughtful review.
>
> > Please explain the reason behind selecting 2-hop ego networks for subgraph construction. Are there specific advantages of this approach compared to other subgraph selection strategies?
>
> The selection of 2-hop ego networks for the subgraph construction was made in order to capture local structural patterns and maintain computational efficiency.
> Proteins exhibit hierarchical organization, where local motifs and secondary structures (e.g., alpha-helices and beta-sheets) are crucial for their function. A 2-hop ego network centered around a given amino acid can capture local structural motifs relevant to the function of the protein and geometric relationships beyond immediate neighbors.
> Using a 1-hop ego network would restrict information flow to only directly connected residues, missing interactions that span slightly larger structural motifs. On the other hand, larger-hop subgraphs (e.g., 3-hop or beyond) could include too much global information, making it harder for the model to disentangle local effects from the broader protein topology.
> Moreover, extracting 2-hop ego networks ensures a manageable subgraph size, preventing computational overhead while still capturing useful context. The extraction of larger subgraphs (e.g., k-hop with large k) can lead to increased memory and compute requirements, therefore it can not scale effectively.
> In contrast, selecting only 1-hop ego networks would result in subgraphs that may be too small to capture meaningful geometric patterns.
> Based on your insightful comment, we have further run experiments with 1-hop ego network selection and random subgraph selection, where we randomly sample $20$ nodes.
> We provide the results in Table 4 in the revised manuscript, which is also attached to this comment.
> Our experiments indicated that using 2-hop ego networks consistently yielded better downstream performance compared to 1-hop and random subgraphs.
>
> | **Model**           | **Subgraph Extraction Method** | **React**         | **Fold**          | **Super-Family**   | **Family**         |
> |---------------------|--------------------------------|-------------------|-------------------|--------------------|--------------------|
> | **GCN Pretrained**     | 2-hop Ego Networks             | 47.46 ± 0.90      | **12.90 ± 0.10**  | **11.81 ± 0.50**   | 58.40 ± 4.20       |
> |     | 1-hop Ego Networks             | **49.86 ± 0.76**  | 11.95 ± 1.74      | 11.35 ± 0.35       | **61.03 ± 2.31**   |
> |       | Random Subgraphs               | 44.60 ± 0.99      | 12.32 ± 0.95      | 11.49 ± 0.32       | 59.87 ± 3.10       |
> | **ProNet Pretrained**   | 2-hop Ego Networks             | **80.61 ± 1.30**  | **50.11 ± 1.00**  | **64.79 ± 2.70**   | **97.88 ± 0.10**   |
> |     | 1-hop Ego Networks             | 75.20 ± 1.10      | 42.27 ± 1.37      | 56.45 ± 0.65       | 96.29 ± 0.38       |
> |     | Random Subgraphs               | 74.11 ± 0.80      | 38.22 ± 1.44      | 50.34 ± 2.04       | 94.37 ± 0.33       |
> | **SchNet Pretrained**   | 2-hop Ego Networks             | **65.03 ± 1.30**  | **23.41 ± 0.20**  | **27.65 ± 1.00**   | **82.62 ± 1.70**   |
> |     | 1-hop Ego Networks             | 61.61 ± 2.62      | 22.05 ± 1.21      | 26.56 ± 1.31       | 80.78 ± 0.98       |
> |     | Random Subgraphs               | 57.77 ± 2.71      | 23.30 ± 1.12      | 26.39 ± 0.49       | 79.10 ± 0.83       |

---

> > ### Author Response · Authors · 2025-03-05
> > **Response 2/n**
> >
> > > Please provide an explanation for the decision to treat all amino acids equally in the model. How does this simplification impact the model’s ability to capture the unique biochemical properties of individual amino acids?
> >
> > In our model, centroid computations are performed without explicitly weighting individual amino acids, but it is important to note that amino acid identity is still preserved in the node features. This means that while the centroid calculation itself remains uniform, the model retains the ability to learn and differentiate amino acid properties through its graph neural network layers.
> > The primary reason for treating all amino acids equally in centroid computation is to ensure that the model captures purely geometric patterns without introducing biases that could negatively impact generalization across diverse proteins. While amino acid-specific biochemical properties are crucial for function, protein structures often share common spatial motifs across different functional classes, even when composed of different residues. By maintaining a uniform centroid calculation, our model remains agnostic to sequence-specific biases and learns geometric representations that are more transferable across datasets and tasks.
> > Furthermore, incorporating task-dependent amino acid weighting could make the method less adaptable to new datasets with varying biochemical properties, as the optimal weighting strategy may differ depending on the specific application (e.g., protein folding vs. ligand binding). Additionally, designing such weighting mechanisms would introduce manual effort and domain-specific tuning, reducing the broad applicability of our approach.
> > However, we emphasize that while the centroid calculation is uniform, the model is still capable of capturing biochemical differences between amino acids through the node-level features, allowing it to learn residue-specific interactions and properties dynamically during training.
> >
> >
> > > If possible, please compare the proposed pretraining method to other state-of-the-art pretraining techniques, such as contrastive learning or hybrid strategies.
> >
> > We appreciate the reviewer's suggestion and, in response, have added a comparison with Infograph[1], a well-known contrastive learning approach for graph representation learning. Specifically, we pretrained different models using InfoGraph in the same pretraining dataset.
> > Our results show that while Infograph achieves competitive performance, our subgraph distance pretraining method consistently outperforms Infograph across multiple metrics.
> > We believe this performance gap stems from a key limitation of many contrastive learning approaches, as they typically rely on generating augmentations by modifying the graph, such as removing edges or nodes or perturbing node features. In the context of proteins, however, even a minor change in an amino acid can have a substantial impact on protein function.
> > Thus, augmentations that disrupt the structure of the protein may lead to information loss. In contrast, our method can avoid the above limitations with the proposed self-supervised objective.
> > We provide the results in the revised Tables 1, 2, and 3. While InfoGraph demonstrates competitive performance, our method achieves better results across most benchmarks. Additionally, our approach is significantly more efficient, running around 30 times faster than InfoGraph, making it more suitable for large pertaining datasets.
> >
> > References:
> >
> > [1] Sun, Fan-Yun, et al. "Infograph: Unsupervised and semi-supervised graph-level representation learning via mutual information maximization." In International Conference on Learning Representations (ICLR 2020).

---

### Review · Reviewer_E1Gw · 2025-01-24

**Summary Of Contributions:**

The paper proposes an approach for self-supervised pre-training of graph networks on protein 3D structure data. The model is trained to perform a surrogate task - predict the distance between the centroid of a subgraph (part of the molecule) and the centroid of the whole graph (whole molecule). One forward pass through the model is performed on the whole molecule and then the features of multiple subgraphs are taken and used for computing the loss, which makes the method compute-efficient.

The method is evaluated on 3 network architectures (ProNet, SchNet, standard GCN) and on 2 downstream prediction tasks related to protein function prediction. In most settings, the method leads to better results than no pre-training or a simple baseline that predicts the lengths of several masked-out edges.

**Audience:**

Yes

**Broader Impact Concerns:**

No specific concerns. Work on protein function prediction can potentially have a huge impact on drug development and beyond, but can also potentially have adverse applications. Im my opinion, the potential upside outweighs the potential downside.

**Claims And Evidence:**

No

**Requested Changes:**

Address the weaknesses:
1. Compare to the baselines from the literature or argue very convincingly why it's not a good idea or is impossible.
2. Strengthen the edge length prediction baseline, at least try different masking ratios.
3. Fix presentation to fit the results.
4. and 5. Fix the presentation issues.

**Strengths And Weaknesses:**

Pros:
1. Mostly fairly good presentation, the method is quite clearly explained.
2. Good experimental results in the considered settings.
3. The authors release the code.

Cons:
1. I find the evaluation presented in the paper not convincing enough, in relation to the claims made in the paper. The paper states: “absence of a unified approach to effectively leverage unlabeled 3D structures for pretraining deep learning models”, suggesting that the proposed method is the first such method, but there are several pre-training methods listed in the related work section but the paper only compares to a naive custom baseline, not to those existing methods.
2. The edge prediction baseline seems quite weak potentially. The 256 edges per batch number is confusing - it is difficult to understand what it means without knowing how many edges there are in the batch. It’s better to report the fraction of masked edges. Is there evidence that the chosen fraction is optimal? Since masking has been done before (as mentioned in related work), is it possible to also compare to those results directly?
3. Multiple confusing/unsupported statements:
a. “In contrast, our pretraining task does not require multiple views, augmentations, or masking strategies” - I would say cutting out subgraphs can be seen as augmentation and one could do contrastive learning with these same subgraphs and also with one forward pass through the network (train the embeddings of subgraphs to be close with the embedding of the whole graph and embeddings of different proteins to be different).
b. “since Mean Squared Error loss is usually much harder to optimize, we discretize the distances using 10 equal bins and formulate the problem as a classification task” - need to provide a reference or experiments demonstrating this or rephrase.
c. “These methods tend to overlook the spatial relationships and the hierarchical organization within protein structures, as they focus solely on single node or edge masking.” - this is not an inherent problem of masking, one could mask out larger parts of the graph, e.g. subgraphs as used in this paper.
4. Presentation issues:
a. “2-hop ego networks” is jargon-y and may be unclear to readers not working in the specific sub-field, please explain what it is or provide a reference.
b. “ReduceLROnPlateau learning rate scheduler monitoring the validation metric with patience of 5 epochs and reduction of 0.6. We monitor the validation accuracy and perform early stopping with patience of 10 epochs, we report the average and standard deviation over three runs using different seeds.” - Is this based on some common practice? The LR scheduler strikes me as somewhat unusual.
c. In Figures 2 and 3, the legend should be made clearer (what are these models?), and generally the figure looks somewhat unprofessional and should ideally be improved.
d. In Figure 2, the big jump in ca_bb looks unhealthy - did the authors investigate it and try to avoid it? (Lower the learning rate, gradient clipping, etc) It might affect the results.
e. Supplementary tables do not seem mentioned in the text. Generally, it would be good to include more results in the more text, summarizing them somehow if space is an issue.
5. Minor writing comments:
a. “the re- searches have started integrating and encoding” - typo “researches”.
b. “most nearest neighbors” -> “nearest neighbors”.
c. “aminoacid” - sometimes written together, should be separated

---

> ### Author Response · Authors · 2025-03-05
> **Response 1/n**
>
> We would like to sincerely thank the reviewer for the constructive comments and suggestions. Please find below our response to your concerns.
>
> > I find the evaluation presented in the paper not convincing enough, in relation to the claims made in the paper. The paper states: “absence of a unified approach to effectively leverage unlabeled 3D structures for pretraining deep learning models”, suggesting that the proposed method is the first such method, but there are several pre-training methods listed in the related work section but the paper only compares to a naive custom baseline, not to those existing methods.
>
> We would like to thank the reviewer for their suggestion and, in response, have added a comparison with Infograph[1], a well-known contrastive learning approach for graph representation learning. Specifically, we pretrained different models using InfoGraph in the same pretraining dataset. We provide the results in Tables 1, 2, and 3 in the revised manuscript.
> Our results show that while Infograph achieves competitive performance, our subgraph distance pretraining method outperforms Infograph in most cases. Additionally, our approach is significantly more efficient, running around 30 times faster than InfoGraph, making it more suitable for large pertaining datasets.
> Generally, we believe this performance gap stems from a key limitation of many contrastive learning approaches, as they typically rely on generating augmentations by modifying the graph, such as removing edges or nodes or perturbing node features. In the context of proteins, however, even a minor change in an amino acid can have a substantial impact on protein function.
> Thus, augmentations that disrupt the structure of the protein may lead to information loss. In contrast, our method can avoid the above limitations with the proposed self-supervised objective.
>
> > The edge prediction baseline seems quite weak potentially. The 256 edges per batch number is confusing - it is difficult to understand what it means without knowing how many edges there are in the batch. It’s better to report the fraction of masked edges. Is there evidence that the chosen fraction is optimal? Since masking has been done before (as mentioned in related work), is it possible to also compare to those results directly?
>
> We follow prior works from the Protein Workshop, which also used 256 edges per batch, ensuring consistency with established methodologies. However, based on your insightful suggestion, we conducted an ablation study by testing 10% and 20% masking rates to assess the impact of different fractions.
> We present the results in the revised manuscript in Tables 13, 14, and 15 (and also attached in this comment) for the three different featurization schemes (ca_base, ca_angles, ca_bb respectively). We observe that the results are similar for the different masking ratios.
>
> ca_base:
> | **Model (Edge Distance)** | **Masked Edges** | **React** | **Fold** | **Super-Family** | **Family** |
> |---------------------------|------------------|-----------|----------|------------------|------------|
> | **GCN (Edge Distance)**   | 10%              | 39.74     | 11.64    | 10.01            | 53.38      |
> |    | 20%              | 38.35     | 11.37    | 10.72            | 50.32      |
> |    | 256 Fixed        | **43.39** | **12.49**| **11.39**        | **54.88**  |
> | **ProNet (Edge Distance)**| 10%              | 79.58     | 47.68    | **63.91**        | 96.00      |
> | | 20%              | **79.98** | **48.12**| 63.31            | 96.19      |
> |  | 256 Fixed        | 79.14     | 47.40    | 63.13            | **98.07**  |
> | **SchNet (Edge Distance)**| 10%              | 63.48     | 23.21    | **30.02**        | **82.47**  |
> | | 20%              | **64.19** | 22.41    | 27.26            | 81.14      |
> |   | 256 Fixed        | 60.95     | 22.16    | 29.36            | 79.60      |
>
> ca_angles:
>
> | **Model (Edge Distance)**   | **Masked Edges** | **React** | **Fold** | **Super-Family** | **Family** |
> |-----------------------------|------------------|-----------|----------|------------------|------------|
> | **GCN (Edge Distance)**     | 10%              | 67.51     | 23.30    | 28.10            | 82.41      |
> |       | 20%              | 66.40     | 24.14    | 31.87            | 85.72      |
> |    | 256 Fixed        | **69.40** | **24.73**| **33.84**        | **88.71**  |
> | **SchNet (Edge Distance)**  | 10%              | 67.77     | 27.22    | 37.85            | **91.34**  |
> |    | 20%              | 68.73     | **28.26**| **39.58**        | 89.89      |
> |    | 256 Fixed        | **68.81** | 27.89    | 36.19            | 90.21      |

---

> > ### Author Response · Authors · 2025-03-05
> > **Response 2/n**
> >
> > ca_bb:
> > | **Model (Edge Distance)**  | **Masked Edges** | **React** | **Fold** | **Super-Family** | **Family** |
> > |----------------------------|------------------|-----------|----------|------------------|------------|
> > | **GCN (Edge Distance)**    | 10%              | 66.98     | **24.28**| **30.62**        | 86.19      |
> > |    | 20%              | 68.97     | 23.13    | 30.15            | 86.82      |
> > |     | 256 Fixed        | **69.80** | 24.21    | 32.31            | **88.31**  |
> > | **SchNet (Edge Distance)** | 10%              | 73.05     | 29.79    | 35.45            | **90.23**  |
> > |  | 20%              | 69.34     | 23.13    | 30.15            | 86.82      |
> > |   | 256 Fixed        | **73.72** | **31.46**| **38.93**        | 90.12      |
> >
> > > Multiple confusing/unsupported statements:
> > a. “In contrast, our pretraining task does not require multiple views, augmentations, or masking strategies” - I would say cutting out subgraphs can be seen as augmentation and one could do contrastive learning with these same subgraphs and also with one forward pass through the network (train the embeddings of subgraphs to be close with the embedding of the whole graph and embeddings of different proteins to be different).
> > b. “since Mean Squared Error loss is usually much harder to optimize, we discretize the distances using 10 equal bins and formulate the problem as a classification task” - need to provide a reference or experiments demonstrating this or rephrase.
> > c. “These methods tend to overlook the spatial relationships and the hierarchical organization within protein structures, as they focus solely on single node or edge masking.” - this is not an inherent problem of masking, one could mask out larger parts of the graph, e.g. subgraphs as used in this paper.
> >
> >
> > a) While it is true that cutting out subgraphs can be viewed as a form of augmentation, our approach fundamentally differs from typical contrastive learning strategies that rely on multiple augmented views.
> > In our method, subgraphs are extracted and the pretraining objective is formulated as a distance prediction task between the subgraph and the global graph centroid. This differs from contrastive objectives that require explicitly generating multiple views and aligning their embeddings.
> > As mentioned in a previous response, even minor changes in the amino-acid sequence or structure can have a substantial impact on protein function. Therefore, aligning the embeddings of those different views could result in uninformative representations.
> > Thus, although subgraph extraction might seem analogous to augmentation, our objective and implementation are designed to preserve biological integrity while being efficient.
> > Based on the reviewer's valuable comment, we have expanded the discussion in the revised manuscript to more thoroughly describe the differences between our approach and contrastive learning methods.
> >
> > b) In many real-world applications, including our own, ground truth distance measurements are subject to annotation noise, inherent measurement uncertainty, and prediction errors.
> > Discretizing these continuous distances into bins allows our model to predict an interval rather than an exact numerical value, which in turn makes it less sensitive to minor deviations that could lead to large penalties in a regression framework.
> > Additionally, using a cross-entropy loss for the classification task yields smoother gradients and more stable convergence during training. We provide additional experimental results, which demonstrate that the classification approach, based on 10 equal bins, outperforms the regression model.
> > We have incorporated these findings in Table 5 of the revised manuscript (and attached in this comment)  and provided a more detailed discussion, with relevant references[2,3].
> >
> > | **Model**           | **Pretraining Objective** | **React**        | **Fold**         | **Super-Family**   | **Family**         |
> > |---------------------|---------------------------|------------------|------------------|--------------------|--------------------|
> > | **GCN Pretrained**      | Regression                | 45.54 ± 3.2      | 12.12 ± 1.8      | 11.32 ± 0.8        | 58.22 ± 1.1        |
> > |      | Classification        | **47.46 ± 0.9**  | **12.90 ± 0.1**  | **11.81 ± 0.5**    | **58.40 ± 4.2**    |
> > | **ProNet Pretrained**   | Regression                | 78.76 ± 1.2      | 47.80 ± 0.01     | 62.45 ± 2.05       | 98.19 ± 0.14       |
> > |    | Classification       | **80.61 ± 1.3**  | **50.11 ± 1.0**  | **64.79 ± 2.7**    | **97.88 ± 0.1**    |
> > | **SchNet Pretrained**   | Regression                | 55.31 ± 1.64     | 22.07 ± 1.78     | 24.95 ± 0.37       | 79.93 ± 0.11       |
> > |     | Classification        | **65.03 ± 1.3**  | **23.41 ± 0.2**  | **27.65 ± 1.0**    | **82.62 ± 1.7**    |

---

> > > ### Author Response · Authors · 2025-03-05
> > > **Response 3/n**
> > >
> > > > c. “These methods tend to overlook the spatial relationships and the hierarchical organization within protein structures, as they focus solely on single node or edge masking.” - this is not an inherent problem of masking, one could mask out larger parts of the graph, e.g. subgraphs as used in this paper
> > >
> > > c) While it is true that masking subgraphs can, in principle, capture spatial and hierarchical information, our approach fundamentally differs in its objective formulation. Instead of masking and reconstructing missing parts of the graph, which still relies on inferring missing data, our method directly predicts the Euclidean distance between the centroid of a subgraph and the global protein centroid. This direct distance prediction provides an explicit supervisory signal that targets the geometric and hierarchical organization inherent in protein structures without the need for reconstructing masked regions.
> > >
> > > > Q4: Presentation issues:
> > > a. “2-hop ego networks” is jargon-y and may be unclear to readers not working in the specific sub-field, please explain what it is or provide a reference.
> > > b. “ReduceLROnPlateau learning rate scheduler monitoring the validation metric with patience of 5 epochs and reduction of 0.6. We monitor the validation accuracy and perform early stopping with patience of 10 epochs, we report the average and standard deviation over three runs using different seeds.” - Is this based on some common practice? The LR scheduler strikes me as somewhat unusual.
> > > c. In Figures 2 and 3, the legend should be made clearer (what are these models?), and generally the figure looks somewhat unprofessional and should ideally be improved.
> > > d. In Figure 2, the big jump in ca_bb looks unhealthy - did the authors investigate it and try to avoid it? (Lower the learning rate, gradient clipping, etc) It might affect the results.
> > > e. Supplementary tables do not seem mentioned in the text. Generally, it would be good to include more results in the more text, summarizing them somehow if space is an issue.
> > >
> > > a) A 2-hop ego network is a subgraph centered around a chosen node (the ego node) that includes both its immediate neighbors (1-hop) and their direct neighbors (2-hop). Based on your suggestion, we have provided a formal definition in the revised manuscript.
> > >
> > > b) The ReduceLROnPlateau learning rate scheduler is commonly used in deep learning to adaptively reduce the learning rate when validation performance plateaus, preventing overfitting and improving convergence stability.
> > > We used the standard hyperparams for the learning rate scheduler from ProteinWorkshop[4] library, which includes protein structure modeling tasks.
> > >
> > > c) Thank you for pointing this out. The models correspond to GCN and SchNet with the three featurisation schemes that we have used across the manuscript, specifically ca\_base uses one-hot encoding of the amino acid type for each node; ca\_angles added 16-dimensional positional encoding and $\kappa, \alpha \in \mathbb{R}^4$ the virtual torsion and bond angles defined over C$\alpha$ atoms; ca\_bb added $\phi, \psi, \omega \in \mathbb{R}^6$ which correspond to the backbone dihedral angles.
> > > We have redesigned the plots with better graphics and clearer captions and legends. We provide the updated figures in the revised manuscript.
> > >
> > > d) Thank you for pointing this out. We experimented with lowering the initial learning rate. While these modifications slightly smoothed the training curve, they did not substantially change the overall convergence trend or the final performance metrics.
> > >
> > > e) We have now added references to the supplementary results in the revised manuscript.
> > >
> > > > Q5: Minor writing comments:
> > > a. “the re- searches have started integrating and encoding” - typo “researches”.
> > > b. “most nearest neighbors” -> “nearest neighbors”.
> > > c. “aminoacid” - sometimes written together, should be separated
> > >
> > >  We have fixed these points in the revised manuscript.
> > >
> > > We appreciate the reviewer's detailed comments and suggestions. We believe that they have significantly improved the clarity and quality of our manuscript.
> > >
> > > References:
> > >
> > > [1] Sun, Fan-Yun, et al. "Infograph: Unsupervised and semi-supervised graph-level representation learning via mutual information maximization." In International Conference on Learning Representations (ICLR 2020).
> > >
> > > [2] Van Den Oord, Aäron, Nal Kalchbrenner, and Koray Kavukcuoglu. "Pixel recurrent neural networks." International conference on machine learning. PMLR, 2016.
> > >
> > > [3] Xiong, Haipeng, and Angela Yao. "Discrete-constrained regression for local counting models." European Conference on Computer Vision. Cham: Springer Nature Switzerland, 2022.
> > >
> > > [4] Jamasb, Arian R., et al. "Evaluating representation learning on the protein structure universe." In International Conference on Learning Representations (ICLR 2024).

---

### Review · Reviewer_Z7Ls · 2025-02-21

**Summary Of Contributions:**

The paper studies a geometric pretraining method on 3D protein structures. This is done by predicting the distances between local centroids of protein subgraphs and the global centroid of the entire protein. The consideration of such fine-grained and local structures enable the 3D GNN to learn discriminative graph representation.

**Audience:**

Yes

**Claims And Evidence:**

No

**Requested Changes:**

See the weakness section.

**Strengths And Weaknesses:**

Strenghts:

- The paper is generally easy to follow and clearly presented.

- The idea of predicting the distances between local centroids of protein subgraphs and the global centroid of the entire protein is interesting. I find it convincing that this objective can well preserve the relationla information of the protein structures.

- Given the simplicity of the self-supervised objective, I think this pretraining method is scalable and may be practically useful.

Weaknesses:

- There are quite a few design choices in the paper. I think the authors did a poor job in studying these design choices.

- I believe that the subgraph selection plays a crucial role in the proposed method. Why choosing 2-hop ego networks as subgraphs? I think this is a critical design choice that should be well justified and ablated. The authors should provide ablation studies for k-hop ego networks to see how k affects the performance. Moreover, is there other subgraph selection methods? Could you make k a random variable that you can sample form some distribution? There are actually a lot of things that can be studied in order to understand how the subgraph selection impacts the final performance.

- Another design choice is the use of centroid. There are many other intergration of the nodes in a graph, say medoid, Arithmetic–geometric mean, geometric mean, etc. It will be nice to conduct experiments on them and see how it affects the final performance.

- Why discretizing the distances using 10 equal bins and then formulating it as a classification problem? How about simply making it a regression problem? I think the authors need to provide justification and comparative experiments on this design choice.

- The experiment section falls short at the comparison to existing baselines. I find the current baselines used in the paper quite weak and limited. The paper only focuses on edge distance prediction as the primary baseline but lacks comparisons with other state-of-the-art geometric pretraining methods, say contrastive learning or rotation-equivariant tasks.

---

> ### Author Response · Authors · 2025-03-05
> **Response 1/n**
>
> We wholeheartedly thank you for your valuable comments and their clear assessment.
>
> > I believe that the subgraph selection plays a crucial role in the proposed method. Why choosing 2-hop ego networks as subgraphs? I think this is a critical design choice that should be well justified and ablated. The authors should provide ablation studies for k-hop ego networks to see how k affects the performance. Moreover, is there other subgraph selection methods? Could you make k a random variable that you can sample form some distribution? There are actually a lot of things that can be studied in order to understand how the subgraph selection impacts the final performance.
>
> Thank you very much for your insightful comment. Indeed, the subgraph selection has a crucial role in our method.
> The selection of 2-hop ego networks for the subgraph construction was made in order to capture local structural patterns and maintain computational efficiency.
> Proteins exhibit hierarchical organization, where local motifs and secondary structures (e.g., alpha-helices and beta-sheets) are crucial for their function. A 2-hop ego network centered around a given amino acid can capture local structural motifs relevant to the function of the protein and geometric relationships beyond immediate neighbors.
> Using a 1-hop ego network would restrict information flow to only directly connected residues, missing interactions that span slightly larger structural motifs. On the other hand, larger-hop subgraphs (e.g., 3-hop or beyond) could include too much global information, making it harder for the model to disentangle local effects from the broader protein topology.
> Moreover, extracting 2-hop ego networks ensures a manageable subgraph size, preventing computational overhead while still capturing useful context. The extraction of larger subgraphs (e.g., k-hop with large $k$) can lead to increased memory and compute requirements, therefore it can not scale effectively.
> In contrast, selecting only 1-hop ego networks would result in subgraphs that may be too small to capture meaningful geometric patterns.
> Based on your insightful comment, we have further run experiments with 1-hop ego network selection and random subgraph selection, where we randomly sample $20$ nodes.
> We have incorporated the additional results in Table $4$ in the revised manuscript, which is also attached to this comment.
> Our experiments indicated that using 2-hop ego networks consistently yielded better downstream performance compared to 1-hop and random subgraphs.
>
> | **Model**            | **Subgraph Extraction Method** | **React**         | **Fold**          | **Super-Family**    | **Family**         |
> |----------------------|--------------------------------|-------------------|-------------------|---------------------|--------------------|
> | **GCN Pretrained**   | 2-hop Ego Networks             | 47.46 ± 0.90      | **12.90 ± 0.10**  | **11.81 ± 0.50**    | 58.40 ± 4.20       |
> |                      | 1-hop Ego Networks             | **49.86 ± 0.76**  | 11.95 ± 1.74      | 11.35 ± 0.35        | **61.03 ± 2.31**   |
> |                      | Random Subgraphs               | 44.60 ± 0.99      | 12.32 ± 0.95      | 11.49 ± 0.32        | 59.87 ± 3.10       |
> | **ProNet Pretrained**| 2-hop Ego Networks             | **80.61 ± 1.30**  | **50.11 ± 1.00**  | **64.79 ± 2.70**    | **97.88 ± 0.10**   |
> |                      | 1-hop Ego Networks             | 75.20 ± 1.10      | 42.27 ± 1.37      | 56.45 ± 0.65        | 96.29 ± 0.38       |
> |                      | Random Subgraphs               | 74.11 ± 0.80      | 38.22 ± 1.44      | 50.34 ± 2.04        | 94.37 ± 0.33       |
> | **SchNet Pretrained**| 2-hop Ego Networks             | **65.03 ± 1.30**  | **23.41 ± 0.20**  | **27.65 ± 1.00**    | **82.62 ± 1.70**   |
> |                      | 1-hop Ego Networks             | 61.61 ± 2.62      | 22.05 ± 1.21      | 26.56 ± 1.31        | 80.78 ± 0.98       |
> |                      | Random Subgraphs               | 57.77 ± 2.71      | 23.30 ± 1.12      | 26.39 ± 0.49        | 79.10 ± 0.83       |

---

> ### Author Response · Authors · 2025-03-05
> **Response 2/n**
>
> > Another design choice is the use of centroid. There are many other intergration of the nodes in a graph, say medoid, Arithmetic–geometric mean, geometric mean, etc. It will be nice to conduct experiments on them and see how it affects the final performance.
>
> We used the arithmetic mean to compute the centroid due to its simplicity and computational efficiency. We acknowledge that experiments comparing different centroid calculation strategies could be valuable, and based on your valuable suggestion, we have included additional results with median and geometric mean in Table 6 in the revised manuscript and attached in this comment. We observe that in most cases, arithmetic mean outperforms the other 2 methods, however, their performance gap is not big.
>
> | **Model**            | **Centroid Method** | **React**         | **Fold**          | **Super-Family**    | **Family**         |
> |----------------------|---------------------|-------------------|-------------------|---------------------|--------------------|
> | **GCN Pretrained**   | Arithmetic Mean     | **47.46 ± 0.9**   | **12.90 ± 0.1**   | 11.81 ± 0.5         | 58.40 ± 4.2        |
> |                      | Geometric Mean      | 45.49 ± 1.6       | 12.36 ± 0.8       | 11.75 ± 0.4         | 59.12 ± 4.3        |
> |                      | Median              | 46.38 ± 1.2       | 12.57 ± 0.3       | **12.39 ± 1.1**     | **61.79 ± 1.8**    |
> | **ProNet Pretrained**| Arithmetic Mean     | **80.61 ± 1.3**   | **50.11 ± 1.0**   | **64.79 ± 2.7**     | **97.88 ± 0.1**    |
> |                      | Geometric Mean      | 74.33 ± 0.9       | 48.63 ± 0.8       | 64.63 ± 2.1         | 97.78 ± 0.3        |
> |                      | Median              | 74.56 ± 1.2       | 46.78 ± 1.1       | 60.52 ± 1.8         | 95.52 ± 0.3        |
> | **SchNet Pretrained**| Arithmetic Mean     | **65.03 ± 1.3**   | 23.41 ± 0.2       | **27.65 ± 1.0**     | **82.62 ± 1.7**    |
> |                      | Geometric Mean      | 58.26 ± 0.8       | **23.87 ± 0.7**   | 25.81 ± 0.3         | 80.73 ± 1.2        |
> |                      | Median              | 59.59 ± 1.4       | 22.04 ± 0.7       | 26.50 ± 0.7         | 80.93 ± 0.9        |
>
>
> > Why discretizing the distances using 10 equal bins and then formulating it as a classification problem? How about simply making it a regression problem? I think the authors need to provide justification and comparative experiments on this design choice.
>
> We thank the reviewer for highlighting this important point. In many real-world applications, including our own, ground truth distance measurements are subject to annotation noise, inherent measurement uncertainty, and prediction errors. Discretizing these continuous distances into bins allows our model to predict an interval rather than an exact numerical value, which in turn makes it less sensitive to minor deviations that could lead to large penalties in a regression framework.
> Additionally, using a cross-entropy loss for the classification task yields smoother gradients and more stable convergence during training. We provide additional experimental results that demonstrate that the classification approach, based on 10 equal bins, outperforms the regression model. We have incorporated these findings in Table 5 of the revised manuscript (also attached in this comment) and provided a more detailed discussion with relevant references on this topic[2,3].
>
> | **Model**           | **Pretraining Objective** | **React**        | **Fold**         | **Super-Family**   | **Family**         |
> |---------------------|---------------------------|------------------|------------------|--------------------|--------------------|
> | **GCN Pretrained**      | Regression                | 45.54 ± 3.2      | 12.12 ± 1.8      | 11.32 ± 0.8        | 58.22 ± 1.1        |
> |      | Classification        | **47.46 ± 0.9**  | **12.90 ± 0.1**  | **11.81 ± 0.5**    | **58.40 ± 4.2**    |
> | **ProNet Pretrained**   | Regression                | 78.76 ± 1.2      | 47.80 ± 0.01     | 62.45 ± 2.05       | 98.19 ± 0.14       |
> |    | Classification       | **80.61 ± 1.3**  | **50.11 ± 1.0**  | **64.79 ± 2.7**    | **97.88 ± 0.1**    |
> | **SchNet Pretrained**   | Regression                | 55.31 ± 1.64     | 22.07 ± 1.78     | 24.95 ± 0.37       | 79.93 ± 0.11       |
> |     | Classification        | **65.03 ± 1.3**  | **23.41 ± 0.2**  | **27.65 ± 1.0**    | **82.62 ± 1.7**    |

---

> > ### Author Response · Authors · 2025-03-05
> > **Response 3/3**
> >
> > > The experiment section falls short at the comparison to existing baselines. I find the current baselines used in the paper quite weak and limited. The paper only focuses on edge distance prediction as the primary baseline but lacks comparisons with other state-of-the-art geometric pretraining methods, say contrastive learning or rotation-equivariant tasks.
> >
> > We appreciate the reviewer's suggestion and, in response, have added a comparison with Infograph[1], a well-known contrastive learning approach for graph representation learning. Specifically, we pretrained different models using InfoGraph in the same pretraining dataset. We provide the results in Tables 1, 2, and 3 in the revised manuscript.
> > Our results show that while Infograph achieves competitive performance, our subgraph distance pretraining method outperforms Infograph in most cases. Additionally, our approach is significantly more efficient, running 30 times faster than InfoGraph, making it more suitable for large pertaining datasets.
> > Generally, we believe this performance gap stems from a key limitation of many contrastive learning approaches, as they typically rely on generating augmentations by modifying the graph, such as removing edges or nodes or perturbing node features. In the context of proteins, however, even a minor change in an amino acid can have a substantial impact on protein function.
> > Thus, augmentations that disrupt the structure of the protein may lead to information loss. In contrast, our method can avoid the above limitations with the proposed self-supervised objective.
> >
> > References:
> >
> > [1] Sun, Fan-Yun, et al. "Infograph: Unsupervised and semi-supervised graph-level representation learning via mutual information maximization." In International Conference on Learning Representations (ICLR 2020).
> >
> > [2] Van Den Oord, Aäron, Nal Kalchbrenner, and Koray Kavukcuoglu. "Pixel recurrent neural networks." International conference on machine learning. PMLR, 2016.
> >
> > [3] Xiong, Haipeng, and Angela Yao. "Discrete-constrained regression for local counting models." European Conference on Computer Vision. Cham: Springer Nature Switzerland, 2022.

---

### Author Response · Authors · 2025-03-05
**General Comment**

We sincerely thank the reviewers for their insightful comments and suggestions. Their feedback has been invaluable in refining our manuscript. In response, we have included additional experiments and ablation studies and have clarified several key aspects of our work. Since some reviewers had similar questions, and we have added several ablation studies, we further summarize our main new results below :

**Ablation on Subgraph Extraction Methods:**

We conducted extensive ablation studies comparing different subgraph extraction methods, including 1-hop ego networks, 2-hop ego networks, and random subgraph sampling . Our experiments, reported in Table 4 of the revised manuscript, consistently show that 2-hop ego networks yield superior downstream performance.

**Ablation on Regression vs Classification:**

We compared a regression-based distance prediction approach with our classification-based formulation (discretizing distances into 10 equal bins). The experimental results in Table 5 demonstrate that the classification approach yields better performance, justifying our design choice.

**Ablation on Centroid Computation Methods:**

To evaluate the impact of different centroid computation methods, we compared arithmetic mean, geometric mean, and median for computing the protein centroid. As presented in Table 6, our results indicate that while the arithmetic mean generally outperforms the other methods, the differences in performance are modest, suggesting that our approach is robust to the choice of centroid calculation.

**Comparison with Contrastive Learning Baseline:**

In response to the request for a broader baseline comparison, we evaluated Infograph [1], a well-known contrastive learning method. Our results indicate that while Infograph achieves competitive performance, our subgraph distance pretraining method not only outperforms it across most benchmarks but is also more efficient.  We believe this performance gap stems from a key limitation of many contrastive learning approaches, as they typically rely on generating augmentations by modifying the graph, such as removing edges or nodes or perturbing node features. In the context of proteins, however, even a minor change in an amino acid can have a substantial impact on protein function. Thus, augmentations that disrupt the structure of the protein may lead to information loss. In contrast, our method can avoid the above limitations with the proposed self-supervised objective.

[1] Sun, Fan-Yun, et al. "Infograph: Unsupervised and semi-supervised graph-level representation learning via mutual information maximization." In International Conference on Learning Representations (ICLR 2020).

---

### Decision · Action_Editor_ses7 · 2025-04-06

**Recommendation:** Reject

**Comment:**

This paper has some values, but also has significant issues that need to be resolved. (1) The presentation and claims seem to be not very consistent with the experiments like "Despite these advancements, a significant limitation remains in the field: the absence of a unified approach to effectively leverage unlabeled 3D structures for pretraining deep learning models.". (2) The experimental results are not very convincing as compared with baseline methods, not only in terms of performance, but if the improvements are really results of the proposed methods. The reviewers are concerned that there are many factors (Subgraph selection method controlled by k, centroid, discretization into bins) that can contribute to the performance, and given so many experiments have been added during revision, a more systematic study is needed in order to show which components of the methods lead to the observed improvements. Also the method shows mixed results, for example, when compare a relatively standard method like InfoGraph, the results are not consistently better. Also more competitive baseline methods need to be compared, as the currently compared baseline methods are relatively weak.



The authors are encouraged to revise the paper by improving the presentation and experiments.

**Audience:**

AI, machine learning, geometric deep learning, AI for science, drug discovery

**Claims And Evidence:**

This paper proposes methods for pre-training 3D protein rep. learning methods based on subgraphs.

**Resubmission Of Major Revision:**

The authors may consider submitting a major revision at a later time.